# ReClor: A Reading Comprehension Dataset Requiring Logical Reasoning

**Weihao Yu**[*], **Zihang Jiang**[*], **Yanfei Dong** & **Jiashi Feng**
National University of Singapore
weihaoyu6@gmail.com, {jzihang, dyanfei}@u.nus.edu,
elefjia@nus.edu.sg

## Abstract

Recent powerful pre-trained language models have achieved remarkable performance on most of the popular datasets for reading comprehension. It is time to introduce more challenging datasets to push the development of this field towards more comprehensive reasoning of text. In this paper, we introduce a new **Re**ading **C**omprehension dataset requiring **lo**gical **r**easoning (ReClor) extracted from standardized graduate admission examinations. As earlier studies suggest, human-annotated datasets usually contain biases, which are often exploited by models to achieve high accuracy without truly understanding the text. In order to comprehensively evaluate the logical reasoning ability of models on ReClor, we propose to identify biased data points and separate them into EASY set while the rest as HARD set. Empirical results show that state-of-the-art models have an outstanding ability to capture biases contained in the dataset with high accuracy on EASY set. However, they struggle on HARD set with poor performance near that of random guess, indicating more research is needed to essentially enhance the logical reasoning ability of current models.[1]

## 1 Introduction

Machine reading comprehension (MRC) is a fundamental task in Natural Language Processing, which requires models to understand a body of text and answer a particular question related to the context. With success of unsupervised representation learning in NLP, language pre-training based models such as GPT-2 (Radford et al., 2019), BERT (Devlin et al., 2019), XLNet (Yang et al., 2019) and RoBERTa (Liu et al., 2019) have achieved nearly saturated performance on most of the popular MRC datasets (Rajpurkar et al., 2016; Lai et al., 2017; Rajpurkar et al., 2018; Wang et al., 2018). It is time to challenge state-of-the-art models with more difficult reading comprehension tasks and move a step forward to more comprehensive analysis and reasoning over text (Dua et al., 2019).

In natural language understanding, logical reasoning is an important ability to ***examine, analyze and critically evaluate arguments as they occur in ordinary language*** according to the definition from Law School Admission Council (2019a). It is a significant component of human intelligence and is essential in negotiation, debate and writing *etc.* However, existing reading comprehension datasets have none or merely a small amount of data requiring logical reasoning, *e.g.*, 0% in MCTest dataset (Richardson et al., 2013) and 1.2% in SQuAD (Rajpurkar et al., 2016) according to Sugawara & Aizawa (2016). One related task is *natural language inference*, which requires models to label the logical relationships of sentence pairs. However, this task only considers three types of simple logical relationships and only needs reasoning at sentence-level. To push the development of models in logical reasoning from simple logical relationship classification to multiple complicated logical reasoning and from sentence-level to passage-level, it is necessary to introduce a reading comprehension dataset targeting logical reasoning.

A typical example of logical reasoning questions is shown in Table 1. Similar to the format of multiple-choice reading comprehension datasets (Richardson et al., 2013; Lai et al., 2017), it contains a context, a question and four options with only one right answer. To answer the question

---

[*]Equal contribution.
[1]Project page: http://whyu.me/reclor/

in this example, readers need to identify the logical connections between the lines to pinpoint the conflict, then understand each of the options and select an option that solves the conflict. Human minds need extensive training and practice to get used to complex reasoning, and it will take immense efforts for crowdsourcing workers to design such logical reasoning questions. Inspired by the datasets extracted from standardized examinations (Lai et al., 2017; Clark et al., 2018), we build a dataset by selecting such logical reasoning questions from standardized exams such as GMAT [2] and LSAT [3]. We finally collect 6,138 pieces of logical reasoning questions, which constitute a **Re**ading **C**omprehension dataset requiring **lo**gical **r**easoning (**ReClor**).

Human-annotated datasets usually contain biases (Schwartz et al., 2017; Cai et al., 2017; Bugert et al., 2017; Poliak et al., 2018; Gururangan et al., 2018; Zellers et al., 2019), which are often exploited by neural network models as shortcut solutions to achieve high testing accuracy. For data points whose options can be selected correctly without knowing the contexts and questions, we classify them as biased ones. In order to fully assess the logical reasoning ability of the models, we propose to identify the biased data points and group them as EASY set, and put the rest into HARD set. Based on our experiments on these separate sets, we find that even the state-of-the-art models can only perform well on EASY set and struggle on HARD set as shown in Figure 1. This phenomenon shows that current models can well capture the biases in the dataset but lack the ability to understand the text and reason based on connections between the lines. On the other hand, human beings perform similarly on both the EASY and HARD set. It is thus observed that there is still a long way to go to equip models with true logical reasoning ability.

The contributions of our paper are two-fold. First, we introduce ReClor, a new reading comprehension dataset requiring logical reasoning. We use option-only-input baselines trained with different random seeds to identify the data points with biases in the testing set, and group them as EASY set, with the rest as HARD set to facilitate comprehensive evaluation. Second, we evaluate several state-of-the-art models on ReClor and find these pre-trained language models can perform well on EASY set but struggle on the HARD set. This indicates although current models are good at exploiting biases in the dataset, they are far from capable of performing real logical reasoning yet.

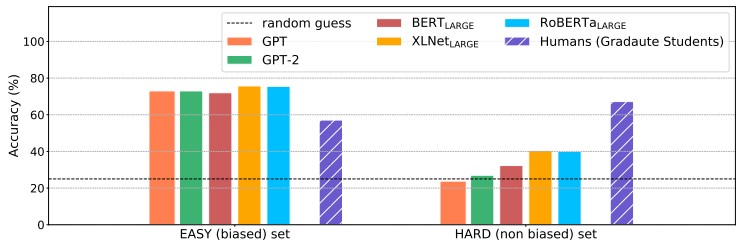

Figure 1: Performance comparison of state-of-the-art models and humans (graduate students) on EASY and HARD set of ReClor testing set.

## 2 RELATED WORK

**Reading Comprehension Datasets.** A variety of reading comprehension datasets have been introduced to promote the development of this field. MCTest (Richardson et al., 2013) is a dataset with 2,000 multiple-choice reading comprehension questions about fictional stories in the format similar to ReClor. Rajpurkar et al. (2016) proposed SQuAD dataset, which contains 107,785 question-answer pairs on 536 Wikipedia articles. The authors manually labeled 192 examples of the dataset and found that the examples mainly require reasoning of lexical or syntactic variation. In an analysis of the above-mentioned datasets, Sugawara & Aizawa (2016) found that none of questions requiring logical reasoning in MCTest dataset (Richardson et al., 2013) and only 1.2% in SQuAD dataset (Rajpurkar et al., 2016). Lai et al. (2017) introduced RACE dataset by collecting the English exams for middle and high school Chinese students in the age range between 12 to 18. They hired crowd workers on Amazon Mechanical Turk to label the reasoning type of 500 samples in the dataset and show that around 70 % of the samples are in the category of word matching, paraphrasing or single-sentence reasoning. To encourage progress on deeper comprehension of language,

---

[2] https://en.wikipedia.org/wiki/Graduate_Management_Admission_Test
[3] https://en.wikipedia.org/wiki/Law_School_Admission_Test

**Context:**
In jurisdictions where use of headlights is optional when visibility is good, drivers who use headlights at all times are less likely to be involved in a collision than are drivers who use headlights only when visibility is poor. Yet Highway Safety Department records show that making use of headlights mandatory at all times does nothing to reduce the overall number of collisions.
**Question:** Which one of the following, if true, most helps to resolve the apparent discrepancy in the information above?
**Options:**
A. In jurisdictions where use of headlights is optional when visibility is good, one driver in four uses headlights for daytime driving in good weather.
B. Only very careful drivers use headlights when their use is not legally required.
C. The jurisdictions where use of headlights is mandatory at all times are those where daytime visibility is frequently poor.
D. A law making use of headlights mandatory at all times is not especially difficult to enforce.
**Answer:** B

Table 1: An example in the ReClor dataset which is modified from the Law School Admission Council (2019b).

more reading comprehension datasets requiring more complicated reasoning types are introduced, such as iterative reasoning about the narrative of a story (Kočiský et al., 2018), multi-hop reasoning across multiple sentences (Khashabi et al., 2018) and multiple documents (Welbl et al., 2018), commonsense knowledge reasoning (Mihaylov et al., 2018; Zhang et al., 2018; Huang et al., 2019) and numerical discrete reasoning over paragraphs (Dua et al., 2019). However, to the best of our knowledge, although there are some datasets targeting logical reasoning in other NLP tasks mentioned in the next section, there is no dataset targeting evaluating logical reasoning in reading comprehension task. This work introduces a new dataset to fill this gap.

**Logical Reasoning in NLP.** There are several tasks and datasets introduced to investigate logical reasoning in NLP. The task of *natural language inference*, also known as *recognizing textual entailment* (Fyodorov et al., 2000; Condoravdi et al., 2003; Bos & Markert, 2005; Dagan et al., 2005; MacCartney & Manning, 2009) requires models to take a pair of sentence as input and classify their relationship types, *i.e.*, ENTAILMENT, NEUTRAL, or CONTRADICTION. SNLI (Bowman et al., 2015) and MultiNLI (Williams et al., 2018) datasets are proposed for this task. However, this task only focuses on sentence-level logical relationship reasoning and the relationships are limited to only a few types. Another task related to logical reasoning in NLP is *argument reasoning comprehension task* introduced by Habernal et al. (2018) with a dataset of this task. Given an argument with a claim and a premise, this task aims to select the correct implicit warrant from two options. Although the task is on passage-level logical reasoning, it is limited to only one logical reasoning type, *i.e.*, identifying warrants. ReClor and the proposed task integrate various logical reasoning types into reading comprehension, with the aim to promote the development of models in logical reasoning not only from sentence-level to passage-level, but also from simple logical reasoning types to the complicated diverse ones.

**Datasets from Examinations.** There have been several datasets extracted from human standardized examinations in NLP, such as RACE dataset (Lai et al., 2017) mentioned above. Besides, NTCIR QA Lab (Shibuki et al., 2014) offers comparative evaluation for solving real-world university entrance exam questions; The dataset of CLEF QA Entrance Exams Task (Rodrigo et al., 2015) is extracted from standardized English examinations for university admission in Japan; ARC dataset (Clark et al., 2018) consists of 7,787 science questions targeting student grade level, ranging from 3rd grade to 9th; The dialogue-based multiple-choice reading comprehension dataset DREAM (Sun et al., 2019) contains 10,197 questions for 6,444 multi-turn multi-party dialogues from English language exams that are designed by human experts to assess the comprehension level of Chinese learners of English. Compared with these datasets, ReClor distinguishes itself by targeting logical reasoning.

## 3 ReClor Data Collection and Analysis

### 3.1 Data collection

The format of data in ReClor is similar to other multiple-choice reading comprehension datasets (Richardson et al., 2013; Lai et al., 2017), where a data point contains a context, a question and four

|  | **ReClor** | **DREAM** | **MCTest** | **ARC** | **RACE** |
|---|---|---|---|---|---|
| construction method | exams | exams | crowd-sourcing | exams | exams |
| context type | written text | dialogues | child's stories | - | written text |
| # of options | 4 | 3 | 4 | 4 | 4 |
| # of context | 6,138 | 6,444 | 660 | - | 27,933 |
| # of questions | 6,138 | 10,197 | 2,640 | 7,787 | 97,687 |
| Vocab size | 26,576 | 13,037 | 8,000 | 6,329 | 136,629 |
| Context Len | 73.6 | 85.9 | 210.1 | - | 321.9 |
| Question Len | 17.0 | 8.6 | 7.8 | 20.5 | 10.0 |
| Option Len | 20.6 | 5.3 | 3.4 | 4.2 | 5.3 |

Table 2: Statistics of several multiple-choice MRC datasets.

answer options, among which only one option is right/most suitable. We collect reading comprehension problems that require complicated logical reasoning. However, producing such data requires the ability to perform complex logical reasoning, which makes it hard for crowdsourcing workers to generate such logical questions. Fortunately, we find the reading comprehension problems in some standardized tests, such as GMAT and LSAT, are highly in line with our expectation.

We construct a dataset containing 6,138 logical reasoning questions sourced from open websites and books. In the original problems, there are five answer options in which only one is right. To comply with fair use of law[4], we shuffle the order of answer options and randomly delete one of the wrong options for each data point, which results in four options with one right option and three wrong options. Furthermore, similar to ImageNet dataset[5], ReClor is available for non-commercial research purpose only. We are also hosting a public evaluation server on EvalAI (Yadav et al., 2019) to benchmark progress on Reclor.

## 3.2 DATA ANALYSIS

As mentioned above, we collect 6,138 data points, in which 91.22% are from actual exams of GMAT and LSAT while others are from high-quality practice exams. They are divided into training set, validation set and testing set with 4,638, 500 and 1,000 data points respectively. The overall statistics of ReClor and comparison with other similar multiple-choice MRC datasets are summarized in Table 2. As shown, ReClor is of comparable size and relatively large vocabulary size. Compared with RACE, the length of the context of ReCor is much shorter. In RACE, there are many redundant sentences in context to answer a question. However, in ReClor, every sentence in the context passages is important, which makes this dataset focus on evaluating the logical reasoning ability of models rather than the ability to extract relevant information from a long context. The length of answer options of ReClor is largest among these datasets. We analyze and manually annotate the types of questions on the testing set and group them into 17 categories, whose percentages and descriptions are shown in Table 3. The percentages of different types of questions reflect those in the logical reasoning module of GMAT and LSAT. Some examples of different types of logical reasoning are listed in Figure 2, and more examples are listed in the Appendix C. Taking two examples, we further express how humans would solve such questions in Table 4, showing the challenge of ReClor.

## 3.3 DATA BIASES IN THE DATASET

The dataset is collected from exams devised by experts in logical reasoning, which means it is annotated by humans and may introduce biases in the dataset. Recent studies have shown that models can utilize the biases in a dataset of natural language understanding to perform well on the task without truly understanding the text (Schwartz et al., 2017; Cai et al., 2017; Bugert et al., 2017; Poliak et al., 2018; Gururangan et al., 2018; Zellers et al., 2019). It is necessary to analyze such data biases to help evaluate models. In the ReClor dataset, the common context and question are shared across the four options for each data point, so we focus on the analysis of the difference in lexical choice and sentence length of the right and wrong options without contexts and questions. We first investigate the biases of lexical choice. We lowercase the options and then use WordPiece tokenization (Wu et al., 2016) of $BERT_{BASE}$ (Devlin et al., 2019) to get the tokens. Similar to

---

[4]https://www.copyright.gov/fair-use/more-info.html
[5]http://image-net.org/download-faq

| Type | Description |
|---|---|
| Necessary Assumptions (11.4%) | identify the claim that must be true or is required in order for the argument to work. |
| Sufficient Assumptions (3.0%) | identify a sufficient assumption, that is, an assumption that, if added to the argument, would make it logically valid. |
| Strengthen (9.4%) | identify information that would strengthen an argument |
| Weaken (11.3%) | identify information that would weaken an argument |
| Evaluation (1.3%) | identify information that would be useful to know to evaluate an argument |
| Implication (4.6%) | identify something that follows logically from a set of premises |
| Conclusion/Main Point (3.6%) | identify the conclusion/main point of a line of reasoning |
| Most Strongly Supported (5.6%) | find the choice that is most strongly supported by a stimulus |
| Explain or Resolve (8.4%) | identify information that would explain or resolve a situation |
| Principle (6.5%) | identify the principle, or find a situation that conforms to a principle, or match the principles |
| Dispute (3.0%) | identify or infer an issue in dispute |
| Technique (3.6%) | identify the technique used in the reasoning of an argument |
| Role (3.2%) | describe the individual role that a statement is playing in a larger argument |
| Identify a Flaw (11.7%) | identify a flaw in an argument's reasoning |
| Match Flaws (3.1%) | find a choice containing an argument that exhibits the same flaws as the passage's argument |
| Match the Structure (3.0%) | match the structure of an argument in a choice to the structure of the argument in the passage |
| Others (7.3%) | other types of questions which are not included by the above |

Table 3: The percentage and description of each logical reasoning type. The descriptions are adapted from those specified by Khan Academy (2019).

Poliak et al. (2018), for the tokens in options, we analyze their conditional probability of label $l \in \{\text{right}, \text{wrong}\}$ given by the token $t$ by $p(l|t) = count(t,l)/count(t)$. The larger the correlation score is for a particular token, the more likely it contributes to the prediction of related option. Table 5 reports tokens in training set which occur at least twenty times with the highest scores since many of the tokens with the highest scores are of low frequency. We further analyze the lengths of right and wrong options (Gururangan et al., 2018) in training set. We notice a slight difference in the distribution of sentence length for right and wrong options. The average length for wrong options is around 21.82 whereas that for right options is generally longer with an average length of 23.06.

| Token | Score (%) | Freq |
|---|---|---|
| motive | 65.2 | 23 |
| ##ce | 62.5 | 24 |
| thereby | 56.0 | 25 |
| consequence | 52.4 | 21 |
| warm | 52.4 | 21 |
| interfere | 52.2 | 23 |
| contributes | 52.2 | 23 |
| manufacture | 52.0 | 25 |
| included | 52.0 | 25 |
| preferences | 52.0 | 25 |

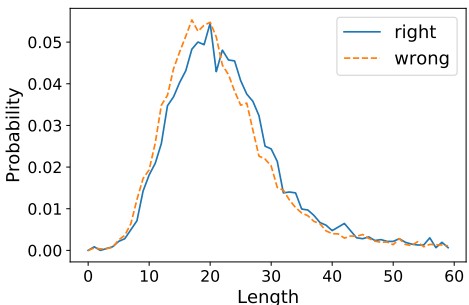

Table 5: Top 10 tokens that correlate to right options with more than 20 occurrences.

Figure 3: The distribution of the option length in ReClor with respect to right and wrong labels.

# 4 EXPERIMENTS

## 4.1 BASELINE MODELS

Many neural network based models such as FastText (Joulin et al., 2017), Bi-LSTM, GPT (Radford et al., 2018), GPT-2 (Radford et al., 2019), BERT (Devlin et al., 2019), XLNet (Yang et al., 2019),

**Context:**
If the purpose of laws is to contribute to people's happiness, we have a basis for criticizing existing laws as well as proposing new laws. Hence, if that is not the purpose, then we have no basis for the evaluation of existing laws, from which we must conclude that existing laws acquire legitimacy simply because they are the laws
**Question:** The reasoning in the argument is flawed in that the argument
**Options:**
A. takes a sufficient condition for a state of affairs to be a necessary condition for it
B. draws a conclusion about how the world actually is on the basis of claims about how it should be
C. infers a causal relationship from the mere presence of a correlation
D. trades on the use of a term in one sense in a premise and in a different sense in the conclusion
**Answer:** A
**Reasoning Process of Humans:**
We may first look at the question to understand the specific task of the question – identify a flaw. We then analyze the argument in the context. The conclusion 'existing laws acquire legitimacy simply because they are the laws.' is based on the argument (*purpose is NOT happiness*) $\rightarrow$ (*NOT basis for criticizing laws*), which is obtained from the first statement: (*purpose is happiness*) $\rightarrow$ (*basis for criticizing laws*). However, we know $\neg A \rightarrow \neg B$ cannot be obtained from $A \rightarrow B$. Therefore, we should choose option A that describes this flaw. The distractors here are different types of reasoning flaws. Prior knowledge of basic logical rules is needed to correctly answer this question.

**Context:**
Psychologist: Phonemic awareness, or the knowledge that spoken language can be broken into component sounds, is essential for learning to read an alphabetic language. But one also needs to learn how sounds are symbolically represented by means of letters; otherwise, phonemic awareness will not translate into the ability to read an alphabetic language. Yet many children who are taught by the whole-language method, which emphasizes the ways words sound, learn to read alphabetic languages.
**Question:** Which one of the following can be properly inferred from the psychologist's statements?
**Options:**
A. The whole-language method invariably succeeds in teaching awareness of how spoken language can be broken into component sounds.
B. Some children who are taught by the whole-language method are not prevented from learning how sounds are represented by means of letters.
C. The whole-language method succeeds in teaching many children how to represent sounds symbolically by means of letters.
D. When the whole-language method succeeds in teaching someone how to represent sounds by means of letters, that person acquires the ability to read an alphabetic language.
**Answer:** B
**Reasoning Process of Humans:**
Looking at the question and we know that it is asking about implication. From the first two sentences in context, we know that there are two necessary conditions to *read an alphabetic language*: *phonemic awareness* and *symbolic letters*. We also learn [(*NOT symbolic letters*) *AND* (*phonemic awareness*)] $\nrightarrow$ *read an alphabetic language* (denoted as Formula 1). The last sentence in the context says that many children are taught by the whole-language method to learn a language. As for option A, from the context, we only know the whole language method works for 'many' children, which cannot be inferred to 'invariably' works. As for option B, combing three sentences in the context, we know that the whole-language method meets the two necessary conditions to learn a language, especially the last sentence mentions 'learn to read alphabetic languages'. Children learn to read alphabetic languages means that they must recognize symbolic letters that represent sound because *symbolic letters* is a necessary condition of *read an alphabetic language*; otherwise, they cannot read because of Formula 1 mentioned above. Therefore, option B is correct. As for option C, from the context we only know the whole-language method teaches *phonemic awareness* and *read an alphabetic language*. *Symbolic letters* may be taught by other methods, so C is wrong. As for D, similar to C, *symbolic letters* may be taught by other methods and we also cannot obtain: *symbolic letters* $\rightarrow$ *read an alphabetic language*.

Table 4: Two examples to show how humans would solve the questions.

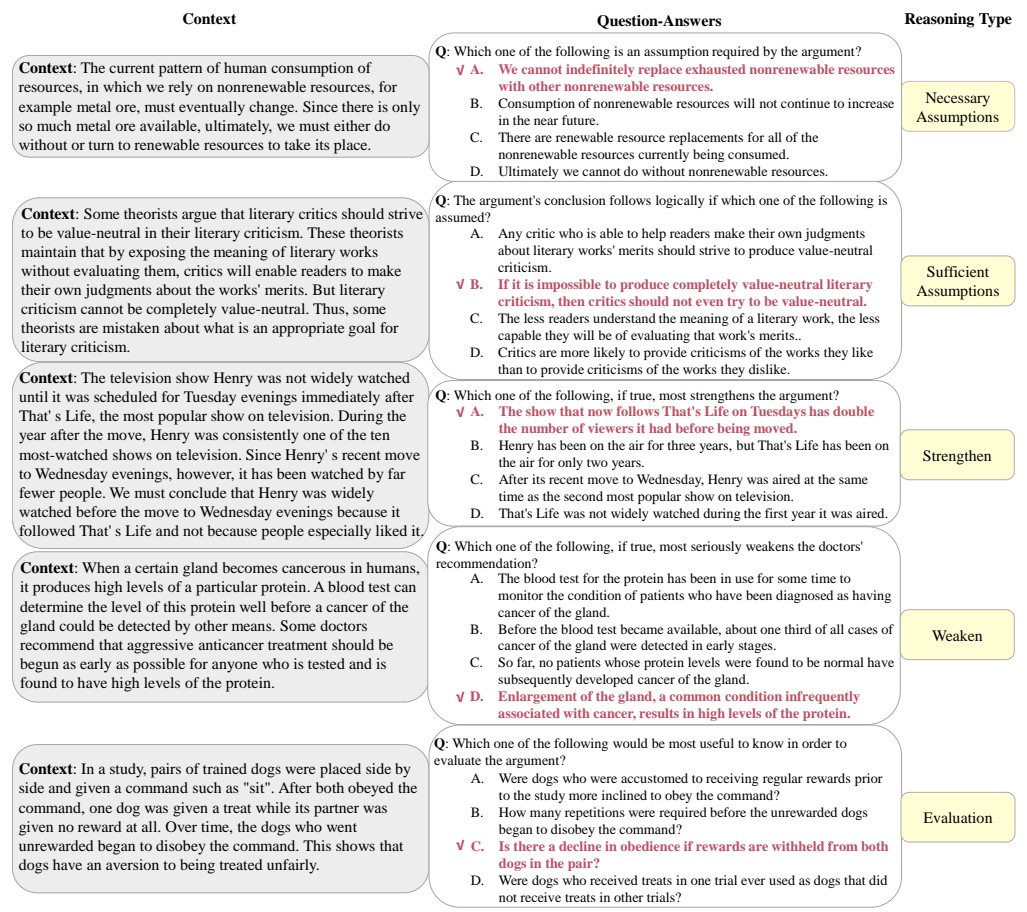

Figure 2: Examples of some question types. The correct options are marked by ✓. More examples are shown in the Appendix C.

RoBERTa (Liu et al., 2019) have achieved impressive results in various NLP tasks. We challenge these neural models with ReClor to investigate how well they can perform. Details of the baseline models and implementation are shown in the Appendix A and B.

## 4.2 EXPERIMENTS TO FIND BIASED DATA

As mentioned earlier, biases prevalently exist in human-annotated datasets (Poliak et al., 2018; Gururangan et al., 2018; Zellers et al., 2019; Niven & Kao, 2019), which are often exploited by models to perform well without truly understanding the text. Therefore, it is necessary to find out the biased data points in ReClor in order to evaluate models in a more comprehensive manner (Sugawara et al., 2018). To this end, we feed the five strong baseline models (GPT, GPT-2, BERT$_{\text{BASE}}$, XLNet$_{\text{BASE}}$ and RoBERTa$_{\text{BASE}}$) with **ONLY THE ANSWER OPTIONS** for each problem. In other words, we purposely remove the context and question in the inputs. In this way, we are able to identify those problems that can be answered correctly by merely exploiting the biases in answer options without knowing the relevant context and question. However, the setting of this task is a multiple-choice question with 4 probable options, and even a chance baseline could have 25% probability to get it right. To eliminate the effect of random guess, we set four different random seeds for each model and pick the data points that are predicted correctly in all four cases to form the EASY set. Then, the data points which are predicted correctly by the models at random could be nearly eliminated, since any data point only has a probability of $(25\%)^4 = 0.39\%$ to be guessed right consecutively for four times. Then we unite the sets of data points that are consistently predicted right by each model,

because intuitively different models may learn different biases of the dataset. The above process is formulated as the following expression,

$$
\begin{aligned}
\mathbb{C}_{\text{EASY}} = & (\mathbb{C}_{\text{GPT}}^{\text{seed}_1} \cap \mathbb{C}_{\text{GPT}}^{\text{seed}_2} \cap \mathbb{C}_{\text{GPT}}^{\text{seed}_3} \cap \mathbb{C}_{\text{GPT}}^{\text{seed}_4}) \\
& \cup (\mathbb{C}_{\text{GPT}-2}^{\text{seed}_1} \cap \mathbb{C}_{\text{GPT}-2}^{\text{seed}_2} \cap \mathbb{C}_{\text{GPT}-2}^{\text{seed}_3} \cap \mathbb{C}_{\text{GPT}-2}^{\text{seed}_4}) \\
& \cup (\mathbb{C}_{\text{BERT}}^{\text{seed}_1} \cap \mathbb{C}_{\text{BERT}}^{\text{seed}_2} \cap \mathbb{C}_{\text{BERT}}^{\text{seed}_3} \cap \mathbb{C}_{\text{BERT}}^{\text{seed}_4}) \\
& \cup (\mathbb{C}_{\text{XLNet}}^{\text{seed}_1} \cap \mathbb{C}_{\text{XLNet}}^{\text{seed}_2} \cap \mathbb{C}_{\text{XLNet}}^{\text{seed}_3} \cap \mathbb{C}_{\text{XLNet}}^{\text{seed}_4}) \\
& \cup (\mathbb{C}_{\text{RoBERTa}}^{\text{seed}_1} \cap \mathbb{C}_{\text{RoBERTa}}^{\text{seed}_2} \cap \mathbb{C}_{\text{RoBERTa}}^{\text{seed}_3} \cap \mathbb{C}_{\text{RoBERTa}}^{\text{seed}_4}),
\end{aligned}
\tag{1}
$$

$$
\mathbb{C}_{\text{HARD}} = \mathbb{C}_{\text{TEST}} - \mathbb{C}_{\text{EASY}},
$$

where $\mathbb{C}_{\text{BERT}}^{\text{seed}_1}$ denotes the set of data points which are predicted correctly by $\text{BERT}_{\text{BASE}}$ with seed 1, and similarly for the rest. Table 6 shows the average performance for each model trained with four different random seeds and the number of data points predicted correctly by all of them. Finally, we get 440 data points from the testing set $\mathbb{C}_{\text{TEST}}$ and we denote this subset as EASY set $\mathbb{C}_{\text{EASY}}$ and the other as HARD set $\mathbb{C}_{\text{HARD}}$.

| Model | Val | Test | Number |
|---|---|---|---|
| Chance | 25.0 | 25.0 | 3.9 |
| GPT | 45.8 | 42.2 | 238 |
| GPT-2 | 46.8 | 42.6 | 245 |
| $\text{BERT}_{\text{BASE}}$ | 47.2 | 43.2 | 234 |
| $\text{XLNet}_{\text{BASE}}$ | 47.5 | 43.2 | 225 |
| $\text{RoBERTa}_{\text{BASE}}$ | 48.8 | 41.7 | 200 |
| Union | – | – | 440 |

Table 6: Average accuracy of each model using four different random seeds with only answer options as input, and the number of their common correct predictions.

### 4.3 TRANSFER LEARNING THROUGH FINE-TUNING

Among multiple-choice reading comprehension or QA datasets from exams, although the size of ReClor is comparable to those of ARC (Clark et al., 2018) and DREAM (Sun et al., 2019), it is much smaller than RACE Lai et al. (2017). Recent studies (Min et al., 2017; Howard & Ruder, 2018; Huang et al., 2019; Jin et al., 2019) have shown the effectiveness of pre-training on similar tasks or datasets then fine-tuning on the target dataset for transfer learning. Jin et al. (2019) find that by first training on RACE (Lai et al., 2017) and then further fine-tuning on the target dataset, the performances of $\text{BERT}_{\text{BASE}}$ on multiple-choice dataset MC500 (Richardson et al., 2013) and DREAM (Sun et al., 2019) can significantly boost from 69.5% to 81.2%, and from 63.2% to 70.2%, respectively. However, they also find that the model cannot obtain significant improvement even performs worse if it is first fine-tuned on span-based dataset like SQuAD (Rajpurkar et al., 2016). ReClor is a multiple-choice dataset, so we choose RACE for fine-tuning study.

### 4.4 RESULTS AND ANALYSIS

The performance of all tested models on the ReClor is presented in Table 7. This dataset is built on questions designed for students who apply for admission to graduate schools, thus we randomly choose 100 samples from the testing set and divide them into ten tests, which are distributed to ten different graduate students in a university. We take the average of their scores and present it as the baseline of graduate students. The data of ReClor are carefully chosen and modified from only high-quality questions from standardized graduate entrance exams. We set the ceiling performance to 100% since ambiguous questions are not included in the dataset.

The performance of fastText is better than random guess, showing that word correlation could be used to help improve performance to some extent. It is difficult for Bi-LSTM to converge on this

| Model | Input | RACE | Val | Test | Test-E | Test-H |
|---|---|---|---|---|---|---|
| Chance | (C, Q, A) | | 25.0 | 25.0 | 25.0 | 25.0 |
| fastText | | | 32.0 | 30.8 | 40.2 | 23.4 |
| Bi-LSTM | | | 27.8 | 27.0 | 26.4 | 27.5 |
| GPT | (C, Q, A) | | 47.6 | 45.4 | 73.0 | 23.8 |
| GPT-2 | | | 52.6 | 47.2 | 73.0 | 27.0 |
| BERT$_{\text{BASE}}$ | (C, Q, A) | | 54.6 | 47.3 | 71.6 | 28.2 |
| | (C, Q, A) | ✓ | 55.2 | 49.5 | 68.9 | 34.3 |
| BERT$_{\text{LARGE}}$ | (A) | | 46.4 | 42.4 | 69.3 | 21.3 |
| | (Q, A) | | 48.8 | 43.4 | 72.7 | 20.4 |
| | (C, Q, A) | | 53.8 | 49.8 | 72.0 | 32.3 |
| | (C, Q, A) | ✓ | 55.6 | 54.5 | 73.9 | 39.3 |
| XLNet$_{\text{BASE}}$ | (C, Q, A) | | 55.8 | 50.4 | 75.2 | 30.9 |
| | (C, Q, A) | ✓ | 62.0 | 55.5 | 76.1 | 39.3 |
| XLNet$_{\text{LARGE}}$ | (A) | | 45.0 | 42.9 | 66.1 | 24.6 |
| | (Q, A) | | 47.8 | 43.4 | 68.6 | 23.6 |
| | (C, Q, A) | | 62.0 | 56.0 | 75.7 | 40.5 |
| | (C, Q, A) | ✓ | 70.8 | 62.4 | 77.7 | 50.4 |
| RoBERTa$_{\text{BASE}}$ | (C, Q, A) | | 55.0 | 48.5 | 71.1 | 30.7 |
| | (C, Q, A) | ✓ | 56.8 | 53.0 | 72.5 | 37.7 |
| RoBERTa$_{\text{LARGE}}$ | (A) | | 48.8 | 43.2 | 69.5 | 22.5 |
| | (Q, A) | | 49.8 | 45.8 | 72.0 | 25.2 |
| | (C, Q, A) | | 62.6 | 55.6 | 75.5 | 40.0 |
| | (C, Q, A) | ✓ | 68.0 | 65.1 | 78.9 | 54.3 |
| Graduate Students | (C, Q, A) | | – | 63.0 | 57.1 | 67.2 |
| Ceiling Performance | (C, Q, A) | | – | 100 | 100 | 100 |

Table 7: Accuracy (%) of models and human performance. The column *Input* means whether to input context (C), question (Q) and answer options (A). The RACE column represents whether to first use RACE to fine-tune before training on ReClor.

dataset. Transformer-based pre-training models have relatively good performance, close to the performance of graduate students. However, we find that these models only perform well on EASY set with around 75% accuracy, showing these models have an outstanding ability to capture the biases of the dataset, but they perform poorly on HARD set with only around 30% accuracy. In contrast, humans can still keep good performance on HARD set. We notice the difference in testing accuracy performed by graduate students on EASY and HARD set, but this could be due to the small number of students participated in the experiments. Therefore, we say humans perform relatively consistent on both biased and non-biased dataset.

It is noticed that if the models are first trained on RACE and then fine-tuned on ReClor, they could obtain significant improvement, especially on HARD set. The overall performance of RoBERTa$_{\text{LARGE}}$ is even better than that of graduate students. This similar phenomenon can also be observed on DREAM dataset (Sun et al., 2019) by Jin et al. (2019), which shows the potential of transfer learning for reasoning tasks. However, even after fine-tuning on RACE, the best performance of these strong baselines on HARD set is around 50%, still lower than that of graduate students and far away from ceiling performance.

Experiments in different input settings are also done. Compared with the input setting of answer options only (A), the setting of questions and answer options (Q, A) can not bring significant improvement. This may be because some questions *e.g.*, *Which one of the following is an assumption required by the argument?*, *Which one of the following, if true, most strengthens the argument?* can be used in the same reasoning types of question, which could not offer much information. Further adding context causes significant boost, showing the high informativeness of the context.

We further analyze the model performance with respect to different question types of logical reasoning. Some results are shown in Figure 4 and the full results are shown in Figure 5, 6 and 7 in the Appendix E. Three models of BERT$_{\text{LARGE}}$, XLNet$_{\text{LARGE}}$ and RoBERTa$_{\text{LARGE}}$ perform well on most of types. On HARD set, the three models perform poorly on certain types such as STRENGTHEN, WEAKEN and ROLE which require extensive logical reasoning. However, they perform relatively better on other certain types, such as CONCLUSION/MAIN POINT and MATCH STRUCTURES that

are more straight-forward. For the result of transfer learning, we analyze XLNet$_{LARGE}$ in detail. Though the overall performance is significantly boosted after fine-tuning on RACE first, the histograms in the bottom of Figure 4 show that on EASY set, accuracy of the model with fine-tuning on RACE is similar to that without it among most question types, while on HARD set, significant improvement on some question types is observed, such as CONCLUSION/MAIN POINT and MOST STRONGLY SUPPORTED. This may be because these types require less logical reasoning to some extent compared with other types, and similar question types may also be found in RACE dataset. Thus, the pre-training on RACE helps enhance the ability of logical reasoning especially of relatively simple reasoning types, but more methods are still needed to further enhance the ability especially that of relatively complex reasoning types.

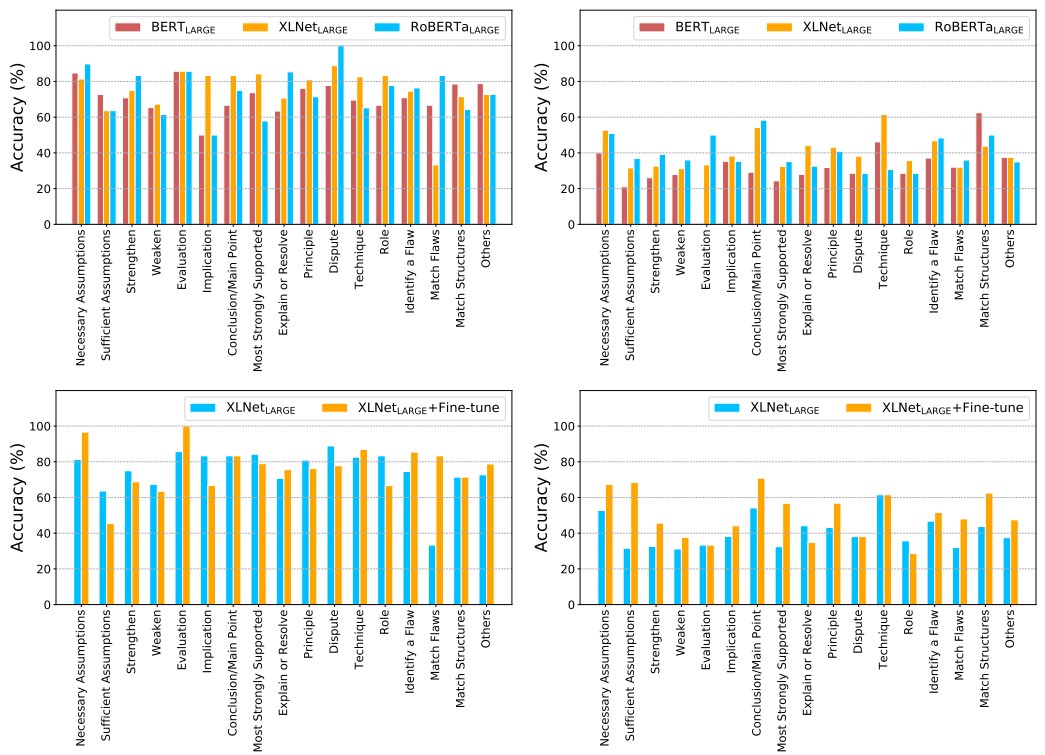

Figure 4: Performance of models on EASY (left) and HARD (right) testing sets and that of models. XLNet$_{LARGE}$ +Fine-Tune means the model is first fine-tuned on RACE before training on ReClor.

## 5 CONCLUSION

In this paper, we introduce ReClor, a reading comprehension dataset requiring logical reasoning, with the aim to push research progress on logical reasoning in NLP forward from sentence-level to passage-level and from simple logical reasoning to multiple complicated one. We propose to identify biased data points and split the testing set into EASY and HARD group for biased and non-biased data separately. We further empirically study the different behaviors of state-of-the-art models on these two testing sets, and find recent powerful transformer-based pre-trained language models have an excellent ability to exploit the biases in the dataset but have difficulty in understanding and reasoning given the non-biased data with low performance close to or slightly better than random guess. These results show there is a long way to equip deep learning models with real logical reasoning abilities. We hope this work would inspire more research in future to adopt similar split technique and evaluation scheme when reporting their model performance. We also show by first fine-tuning on a large-scale dataset RACE then fine-tuning on ReClor, the models could obtain significant improvement, showing the potential of transfer learning to solve reasoning tasks.

ACKNOWLEDGMENTS

We would like to thank the anonymous reviewers for their insightful comments and suggestions; thank Rishabh Jain from Georgia Tech for helping build up the leaderboard of ReClor on EvalAI. Jiashi Feng was partially supported by NUS IDS R-263-000-C67-646, ECRA R-263-000-C87-133, MOE Tier-II R-263-000-D17-112 and AI.SG R-263-000-D97-490. Weihao Yu and Zihang Jiang would like to thank TFRC program for the support of computational resources.

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

# A    BASELINE MODELS

**fastText.**    FastText  (Joulin et al., 2017) models sentences as a bag of n-grams, and tries to predict the probability of each answer being correct independently. We choose the answer with the highest score as the prediction for the multiple-choice setting.

**LSTM sentence encoder.**    A two-layer bi-LSTM is randomly initialized as a sentence encoder with GloVe word embedding (Pennington et al., 2014). With a span of text as input, the last hidden state of the second layer is max-pooled and then fed into a fully-connected layer to compute the output score.

**GPT and GPT-2.**    GPT (Radford et al., 2018) and GPT-2 (Radford et al., 2019) are both transformer (Vaswani et al., 2017) based models which are pre-trained using unsupervised method with a standard language modeling objective. GPT is pre-trained on BooksCorpus; GPT-2 is pre-trained using a larger dataset called WebText. Here we use the smallest model proposed in (Radford et al., 2019) as our GPT-2 baseline. To fine-tune on ReClor, the final hidden vector corresponding to the last input token (`_classify_`) is used as the aggregate representation followed by an extra fully connected layer to compute the score.

**BERT.**    BERT (Devlin et al., 2019) is also a transformer (Vaswani et al., 2017) based model which is trained by using BooksCorpus (Zhu et al., 2015) and English Wikipedia in two unsupervised tasks, i.e., Masked LM (MLM) and Next Sentence Prediction (NSP). During fine-tuning, the final hidden vector corresponding to the first input token (`[CLS]`) is used as the aggregate representation followed by two extra fully connected layers to compute the score.

**XLNet.**    XLNet (Yang et al., 2019) is trained with Permutation Language Modeling and without NSP. In addition, beside BooksCorpus and English Wikipedia used in BERT, it uses Giga5 (Parker et al., 2011), ClueWeb 2012-B (extended from (Callan et al., 2009)), and Common Crawl (com, 2019) for pre-training. We use the final hidden vector corresponding to the last input token `<cls>` as the aggregate representation and introduce two fully connected layers to predict the score.

**RoBERTa.**    RoBERTa (Liu et al., 2019) is an improved pre-training procedure of BERT with training the model longer, with bigger batches over more data and removing NSP objective *etc.*. Extra two fully connected layers are added to transform the final hidden vector of the first input token (`` to the score.

The input format of different models is shown in Table 8.

| Model | Input Format |
|---|---|
| GPT Radford et al. (2018) | `_start_ Context _delimiter_ Question || Option _classify_` |
| GPT-2 Radford et al. (2019) | `_start_ Context _delimiter_ Question || Option _classify_` |
| BERT (Devlin et al., 2019) | `[CLS] Context [SEP] Question || Option [SEP] [PAD]...` |
| XLNet (Yang et al., 2019) | `<pad>...  Context <sep> Question || Option <sep> <cls>` |
| RoBERTa (Liu et al., 2019) | ` Context   Question || Option  <pad>...` |

Table 8: Input formats of different models. `Context`, `Question` and `Option` represent the token sequences of the context, question and option respectively, and `||` denotes concatenation.

# B    IMPLEMENTATION DETAIL

Adam is used by all models. For fastText, we use its python library[6] by converting ReClor to the required form, and keep the default setting of the hyper parameters. For Bi-LSTM, we use a two-layer Bidirectional LSTM with the GloVe 300d word embedding (Pennington et al., 2014) followed by max-pooling and a fully-connected layer. We train the model for 100 epochs using a batch size of 64 and learning rate of $0.1$. A learning rate decay of $0.5$ is also applied every 10 epochs. For pre-training models, we modify the code of Transformers of Hugging Face[7] to implement them on ReClor. We use a batch size of 24 and fine-tune for 10 epochs. The maximum input sequence length for all models is 256. The detailed hyperparameters are shown in Table 9.

---

[6]`https://github.com/facebookresearch/fastText`
[7]`https://github.com/huggingface/transformers`

| HYPERPARAM | GPT | GPT-2 | BERT$_{BASE}$ | BERT$_{LARGE}$ | XLNet$_{BASE}$ | XLNet$_{LARGE}$ | RoBERTa$_{BASE}$ | RoBERTa$_{LARGE}$ |
|---|---|---|---|---|---|---|---|---|
| Learning Rate | 6.25e-5 | 6.25e-5 | 2e-5 | 2e-5 | 2e-5 | 2e-5 | 1e-5 | 1e-5 |
| Batch Size | | | | | 24 | | | |
| Max Seq Length | | | | | 256 | | | |
| Learning Rate Decay | | | | | Linear | | | |
| Number of Epochs | | | | | 10 | | | |
| Warm-up Proportion | | | | | 0.1 | | | |
| Weight Decay | 0.01 | 0.01 | 0.0 | 0.0 | 0.01 | 0.01 | 0.01 | 0.01 |
| Adam Epsilon | 1e-8 | 1e-8 | 1e-6 | 1e-6 | 1e-6 | 1e-6 | 1e-6 | 1e-6 |
| Adam Betas | (0.9, 0.999) | (0.9, 0.999) | (0.9, 0.999) | (0.9, 0.999) | (0.9, 0.999) | (0.9, 0.999) | (0.9, 0.98) | (0.9, 0.98) |
| Clip Grad Norm | | | | | Not | | | |

Table 9: Hyperparameters for finetuning pre-training language models on ReClor

## C  EXAMPLES

| |
|---|
| **Type:**  Necessary Assumptions |
| **Definition:** identify the claim that must be true or is required in order for the argument to work |
| **Context:** |
| Slash-and-burn agriculture involves burning several acres of forest, leaving vegetable ash that provides ample fertilizer for three or four years of bountiful crops. On the cleared land nutrients leach out of the soil, however, and the land becomes too poor to support agriculture. New land is then cleared by burning and the process starts again. Since most farming in the tropics uses this method, forests in this region will eventually be permanently eradicated. |
| **Question:**  The argument depends on the assumption that |
| **Options:** |
| A. forests in the tropics do not regenerate well enough to restore themselves once they have been cleared by the slash-and-burn method |
| B. some other methods of agriculture are not as destructive to the environment in tropical regions as the slash-and-burn method is |
| C. forests in the tropics are naturally deficient in nutrients that are needed to support the growth of plants that are not native to those regions |
| D. slash-and-burn agriculture is particularly suitable for farming in tropical areas |
| **Answer:** A |

Table 10: The definition and an example of the logical reasoning type - Necessary Assumptions

| |
|---|
| **Type:** Sufficient Assumptions |
| **Definition:** identify a sufficient assumption, that is, an assumption that, if added to the argument, would make it logically valid |
| **Context:** |
| Geologist: A new method for forecasting earthquakes has reliably predicted several earthquakes. Unfortunately, this method can predict only that an earthquake will fall somewhere within a range of two and a half points on the Richter scale. Thus, since a difference of two and a half points can be the difference between a marginally perceptible shaking and a quake that causes considerable damage, the new method is unlikely to be useful. |
| **Question:**  Which one of the following, if assumed, enables the geologist's conclusion to be properly inferred? |
| **Options:** |
| A. An earthquake-forecasting method is unlikely to be useful unless its predictions always differentiate earthquakes that are barely noticeable from ones that result in substantial destruction. |
| B. Several well-established methods for forecasting earthquakes can predict within much narrower ranges than two and a half points on the Richter scale. |
| C. Even if an earthquake-forecasting method makes predictions within a very narrow range on the Richter scale, this method is not likely to be useful unless its predictions are reliable. |
| D. An earthquake-forecasting method has not been shown to be useful until it has been used to reliably predict a large number of earthquakes. |
| **Answer:** A |

Table 11: The definition and an example of the logical reasoning type - Sufficient Assumptions

**Type:** Strengthen
**Definition:** identify information that would strengthen an argument
**Context:**
Financial success does not guarantee happiness. This claim is not mere proverbial wisdom but a fact verified by statistics. In a recently concluded survey, only one-third of the respondents who claimed to have achieved financial success reported that they were happy.
**Question:** Which one of the following, if true, most strongly supports the conclusion drawn from the survey results?
**Options:**
A. Most of the respondents who reported they were unhappy were in fact happy.
B. The respondents who reported financial success were, for the most part, financially successful.
C. Many of the respondents who claimed not to have achieved financial success reported that they were happy five years ago.
D. Many of the respondents who failed to report financial success were in fact financially successful.
**Answer:** B

Table 12: The definition and an example of the logical reasoning type - Strengthen

**Type:** Weaken
**Definition:** identify information that would weaken an argument
**Context:**
"DNA fingerprinting" is a recently-introduced biochemical procedure that uses a pattern derived from a person's genetic material to match a suspect's genetic material against that of a specimen from a crime scene. Proponents have claimed astronomically high odds against obtaining a match by chance alone. These odds are based on an assumption that there is independence between the different characteristics represented by a single pattern.
**Question:** Which one of the following, if true, casts the most doubt on the claim of the proponents of DNA fingerprinting?
**Options:**
A. The skill required of laboratory technicians performing the DNA fingerprinting procedure is not extraordinary.
B. There is a generally accepted theoretical basis for interpreting the patterns produced by the procedure.
C. In the whole population there are various different subgroups, within each of which certain sets of genetic characteristics are shared.
D. In the investigation of certain genetic diseases, the techniques used in DNA fingerprinting have traced the transmission of the diseases among the living members of very large families.
**Answer:** C

Table 13: The definition and an example of the logical reasoning type - Weaken

**Type:** Evaluation
**Definition:** identify information that would be useful to know to evaluate an argument
**Context:**
George: Some scientists say that global warming will occur because people are releasing large amounts of carbon dioxide into the atmosphere by burning trees and fossil fuels. We can see, though, that the predicted warming is occurring already. In the middle of last winter, we had a month of springlike weather in our area, and this fall, because of unusually mild temperatures, the leaves on our town's trees were three weeks late in turning color.
**Question:** Which one of the following would it be most relevant to investigate in evaluating the conclusion of George's argument?
**Options:**
A. whether air pollution is causing some trees in the area to lose their leaves
B. what proportion of global emissions of carbon dioxide is due to the burning of trees by humans
C. whether unusually warm weather is occurring elsewhere on the globe more frequently than before
D. when leaves on the trees in the town usually change color
**Answer:** C

Table 14: The definition and an example of the logical reasoning type - Evaluation

| **Type:** Implication |
| --- |
| **Definition:** identify something that follows logically from a set of premises |
| **Context:** |
| To be horrific, a monster must be threatening. Whether or not it presents psychological, moral or social dangers, or triggers enduring infantile fears, if a monster is physically dangerous then it is threatening. In fact, even a physically benign monster is horrific if it inspires revulsion. |
| **Question:** Which one of the following logically follows from the statements above? |
| **Options:** |
| A. Any horror-story monster that is threatening is also horrific. |
| B. If a monster triggers infantile fears but is not physically dangerous, then it is not horrific. |
| C. All monsters that are not physically dangerous, but that are psychologically dangerous and inspire revulsion, are threatening. |
| D. If a monster is both horrific and psychologically threatening, then it does not inspire revulsion. |
| **Answer:** C |

Table 15: The definition and an example of the logical reasoning type - Implication

| **Type:** Conclusion/Main Point |
| --- |
| **Definition:** identify the conclusion/main point of a line of reasoning |
| **Context:** |
| Whether or not one can rightfully call a person's faithfulness a virtue depends in part on the object of that person's faithfulness. Virtues are by definition praiseworthy, which is why no one considers resentment virtuous, even though it is in fact a kind of faithfulness – faithfulness to hatreds or animosities. |
| **Question:** Which one of the following most accurately expresses the overall conclusion drawn in the argument? |
| **Options:** |
| A. The object of a person's faithfulness partially determines whether or not that faithfulness is virtuous. |
| B. Virtuous behavior is praiseworthy by definition. |
| C. Resentment should not be considered a virtuous emotion. |
| D. Behavior that emerges from hatred or animosity cannot be called virtuous. |
| **Answer:** A |

Table 16: The definition and an example of the logical reasoning type - Conclusion/Main Point

| **Type:** Most Strongly Supported |
| --- |
| **Definition:** find the choice that is most strongly supported by a stimulus |
| **Context:** |
| After a nuclear power plant accident, researchers found radioactive isotopes of iodine, tellurium, and cesium-but no heavy isotopes-in the atmosphere downwind. This material came either from spent fuel rods or from the plant's core. Spent fuel rods never contain significant quantities of tellurium isotopes. Radioactive material ejected into the atmosphere directly from the core would include heavy isotopes. After the accident, steam, which may have been in contact with the core, was released from the plant. The core contains iodine, tellurium, and cesium isotopes, which are easily dissolved by steam. |
| **Question:** |
| Of the following statements, which one is most strongly supported by the information above? |
| **Options:** |
| A. The nuclear power plant's spent fuel rods were not damaged. |
| B. Spent fuel rods do not contain heavy isotopes in significant quantities. |
| C. The researchers found some radioactive material from spent fuel rods as well as some material that was ejected into the atmosphere directly from the plant's core. |
| D. The radioactive material detected by the researchers was carried into the atmosphere by the steam that was released from the plant. |
| **Answer:** D |

Table 17: The definition and an example of the logical reasoning type - Most Strongly Supported

**Type:** Explain or Resolve
**Definition:** identify information that would explain or resolve a situation

**Context:**
To reduce the mosquito population in a resort area, hundreds of trees were planted that bear fruit attractive to birds. Over the years, as the trees matured, they attracted a variety of bird species and greatly increased the summer bird population in the area. As expected, the birds ate many mosquitoes. However, the planting of the fruit trees had the very opposite of its intended effect.

**Question:**
Which one of the following, if true, most helps to explain the apparently paradoxical result?

**Options:**
A. Most of the species of birds that were attracted by the trees that were planted did not eat mosquitoes.
B. Increases and decreases in mosquito populations tend to follow a cyclical pattern.
C. The species of birds that were attracted in the greatest number by the fruit of the trees that were planted did not eat mosquitoes.
D. The birds attracted to the area by the trees ate many more insects that prey on mosquitoes than they did mosquitoes.

**Answer:** D

Table 18: The definition and an example of the logical reasoning type - Explain or Resolve

**Type:** Principle
**Definition:** identify the principle, or find a situation that conforms to a principle, or match the principles

**Context:**
Buying elaborate screensavers – programs that put moving images on a computer monitor to prevent damage – can cost a company far more in employee time than it saves in electricity and monitor protection. Employees cannot resist spending time playing with screensavers that flash interesting graphics across their screens.

**Question:**
Which one of the following most closely conforms to the principle illustrated above?

**Options:**
A. An electronic keyboard may be cheaper to buy than a piano but more expensive to repair.
B. An energy-efficient insulation system may cost more up front but will ultimately save money over the life of the house.
C. The time that it takes to have a pizza delivered may be longer than it takes to cook a complete dinner.
D. A complicated hotel security system may cost more in customer goodwill than it saves in losses by theft.

**Answer:** D

Table 19: The definition and an example of the logical reasoning type - Principle

**Type:** Dispute
**Definition:** identify or infer an issue in dispute

**Context:**
Raphaela: Forcing people to help others is morally wrong. Therefore, no government has the right to redistribute resources via taxation. Anyone who wants can help others voluntarily. Edward: Governments do have that right, insofar as they give people the freedom to leave and hence not to live under their authority.

**Question:**
Raphaela and Edward disagree about the truth of which one of the following?

**Options:**
A. Any government that forces people to help others should permit emigration.
B. Any government that permits emigration has the right to redistribute resources via taxation.
C. Any government that redistributes resources via taxation forces people to help others.
D. Every government should allow people to help others voluntarily.

**Answer:** B

Table 20: The definition and an example of the logical reasoning type - Dispute

| |
|---|
| **Type:** Technique |
| **Definition:** identify the technique used in the reasoning of an argument |
| **Context:** Joanna: The only way for a company to be successful, after emerging from bankruptcy, is to produce the same goods or services that it did before going bankrupt. It is futile for such a company to try to learn a whole new business. Ruth: Wrong. The Kelton Company was a major mining operation that went into bankruptcy. On emerging from bankruptcy, Kelton turned its mines into landfills and is presently a highly successful waste-management concern. |
| **Question:** Ruth uses which one of the following argumentative techniques in countering Joanna's argument? |
| **Options:** A. She undermines a claim by showing that it rests on an ambiguity. B. She offers an alternative explanation for a phenomenon. C. She presents a counterexample to a claim. D. She establishes a conclusion by excluding the only plausible alternative to that conclusion. |
| **Answer:** C |

Table 21: The definition and an example of the logical reasoning type - Technique

| |
|---|
| **Type:** Role |
| **Definition:** describe the individual role that a statement is playing in a larger argument |
| **Context:** The position that punishment should be proportional to how serious the offense is but that repeat offenders should receive harsher punishments than first-time offenders is unsustainable. It implies that considerations as remote as what an offender did years ago are relevant to the seriousness of an offense. If such remote considerations were relevant, almost every other consideration would be too. But this would make determining the seriousness of an offense so difficult that it would be impossible to apply the proportionality principle. |
| **Question:** The statement that considerations as remote as what an offender did years ago are relevant to the seriousness of an offense plays which one of the following roles in the argument? |
| **Options:** A. It is an allegedly untenable consequence of a view rejected in the argument's overall conclusion. B. It is a statement the argument provides grounds to accept and from which the overall conclusion is inferred. C. It is the overall conclusion in favor of which the argument offers evidence. D. It is a premise offered in support of an intermediate conclusion of the argument. |
| **Answer:** A |

Table 22: The definition and an example of the logical reasoning type - Role

| |
|---|
| **Type:** Identify a Flaw |
| **Definition:** identify a flaw in an argument's reasoning |
| **Context:** The tidal range at a particular location is the difference in height between high tide and low tide. Tidal studies have shown that one of the greatest tidal ranges in the world is found in the Bay of Fundy and reaches more than seventeen meters. Since the only forces involved in inducing the tides are the sun's and moon's gravity, the magnitudes of tidal ranges also must be explained entirely by gravitational forces. |
| **Question:** Which one of the following most accurately describes a flaw in the reasoning above? |
| **Options:** A. It does not differentiate between the tidal effect of the sun and the tidal effect of the moon. B. It fails to consider that the size of a tidal range could be affected by the conditions in which gravitational forces act. C. It presumes, without providing warrant, that most activity within the world's oceans is a result of an interplay of gravitational forces. D. It gives only one example of a tidal range. |
| **Answer:** B |

Table 23: The definition and an example of the logical reasoning type - Identify a Flaw

| **Type:** Match Flaws |
| --- |
| **Definition:** find a choice containing an argument that exhibits the same flaws as the passage's argument |
| **Context:** |
| The museum' s night security guard maintains that the thieves who stole the portrait did not enter the museum at any point at or above ground level. Therefore, the thieves must have gained access to the museum from below ground level. |
| **Question:** |
| The flawed pattern of reasoning in the argument above is most similar to that in which one of the following? |
| **Options:** |
| A. As had generally been expected, not all questionnaires were sent in by the official deadline. It follows that plans must have been made for the processing of questionnaires received late. |
| B. The store's competitors claim that the store, in selling off the shirts at those prices, neither made any profit nor broke even. Consequently, the store's customers must have been able to buy shirts there at less than the store's cost. |
| C. The product label establishes that this insecticide is safe for both humans and pets. Therefore, the insecticide must also be safe for such wild mammals as deer and rabbits. |
| D. If the census is to be believed, the percentage of men who are married is higher than the percentage of women who are married. Thus, the census must show a higher number of men than of women overall. |
| **Answer:** B |

Table 24: The definition and an example of the logical reasoning type - Match Flaws

| **Type:** Match the Structure |
| --- |
| **Definition:** match the structure of an argument in a choice to the structure of the argument in the passage |
| **Context:** |
| It is an absurd idea that whatever artistic endeavor the government refuses to support it does not allow, as one can see by rephrasing the statement to read: No one is allowed to create art without a government subsidy. |
| **Question:** |
| The pattern of reasoning in which one of the following is most similar to that in the argument above? |
| **Options:** |
| A. The notion that every scientist who has been supported by a government grant will be successful is absurd, as one can see by rewording it:No scientist is allowed to do research without a government grant. |
| B. The notion that every scientist who is supported by a government grant will be successful is absurd, as one can see by rewording it:No scientist lacking governmental support will be successful. |
| C. The claim that any driver who is not arrested does not break the law is absurd, as one can see by rewording it: Every driver who gets arrested has broken the law. |
| D. The claim that any driver who is not arrested does not break the law is absurd, as one can see by rewording it: Every driver who breaks the law gets arrested. |
| **Answer:** D |

Table 25: The definition and an example of the logical reasoning type - Match the Structure

| **Type:** Others |
| --- |
| **Definition:** other types of questions which are not included by the above |
| **Context:** |
| PhishCo runs a number of farms in the arid province of Nufa, depending largely on irrigation. Now, as part of a plan to efficiently increase the farms' total production, it plans to drill down to an aquifer containing warm, slightly salty water that will be used to raise fish in ponds. The water from the ponds will later be used to supplement piped-in irrigation water for PhishCo's vegetable fields, and the ponds and accompanying vegetation should help reduce the heat in the area of the farms. |
| **Question:** |
| Which of the following would, if true, most strongly suggest that the plan, if implemented, would increase the overall efficiency of PhishCo's farms? |
| **Options:** |
| A. Organic waste from fish in the pond water will help to fertilize fields where it is used for irrigation. |
| B. Fish raised on PhishCo's farms are likely to be saleable in the nearest urban areas. |
| C. Ponds will be located on low-lying land now partially occupied by grain crops. |
| D. The government of Nufa will help to arrange loan financing to partially cover the costs of drilling. |
| **Answer:** A |

Table 26: The definition and an example of the logical reasoning type - Others

# D    CONSISTENCY OF DIFFERENT MODELS

| | GPT | GPT-2 | BERT$_{BASE}$ | XLNet$_{BASE}$ | RoBERTa$_{BASE}$ |
|---|---|---|---|---|---|
| GPT | **245** | 164 | 152 | 142 | 116 |
| GPT-2 | | **238** | 151 | 144 | 123 |
| BERT$_{BASE}$ | | | **234** | 138 | 124 |
| XLNet$_{BASE}$ | | | | **225** | 125 |
| RoBERTa$_{BASE}$ | | | | | **200** |

Table 27: Overlap of each pair of models after intersection among 4 random seeds.

# E    RESULTS WITH RESPECT TO DIFFERENT QUESTION TYPES

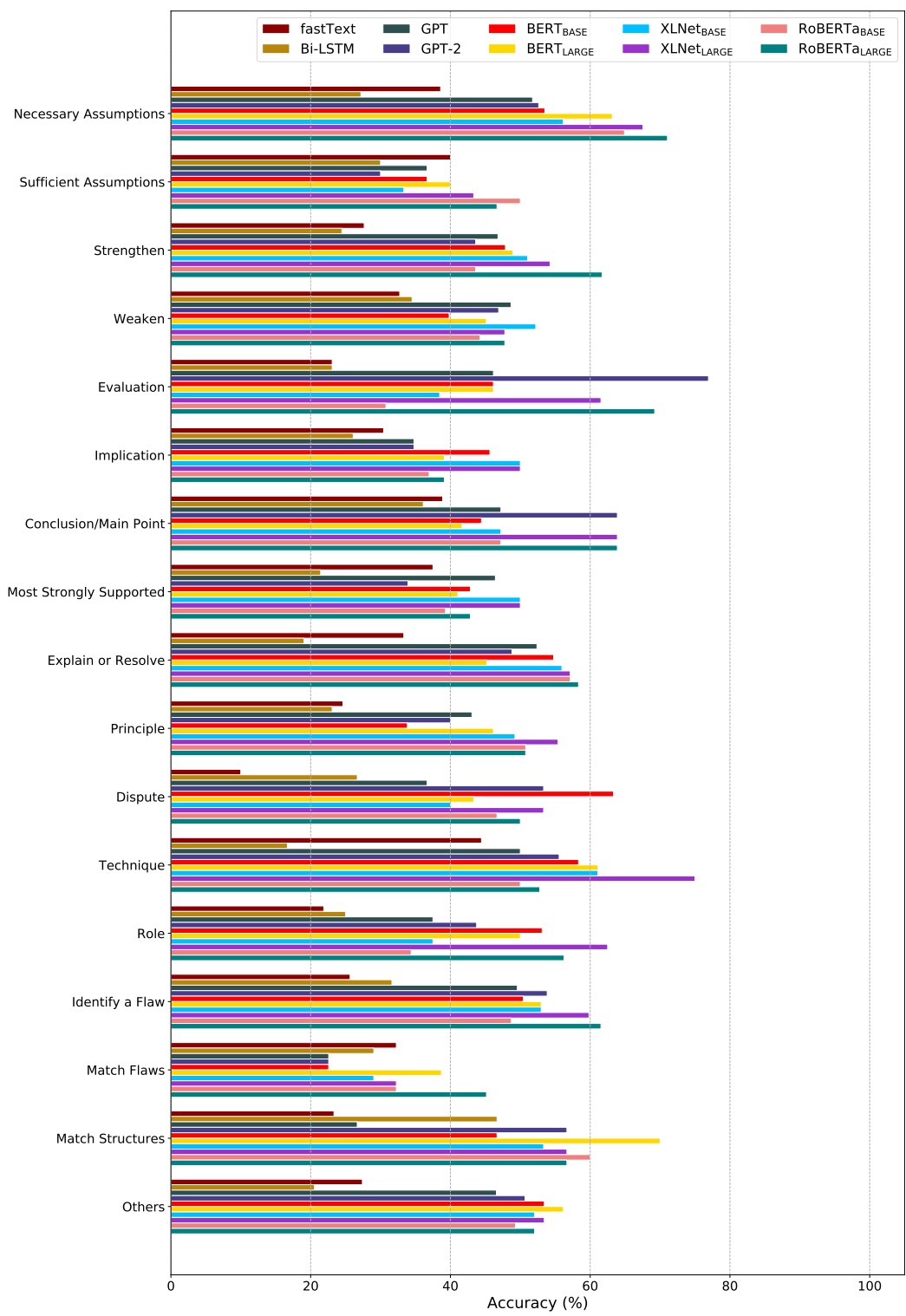

Figure 5: Accuracy of all baseline models on overall testing set

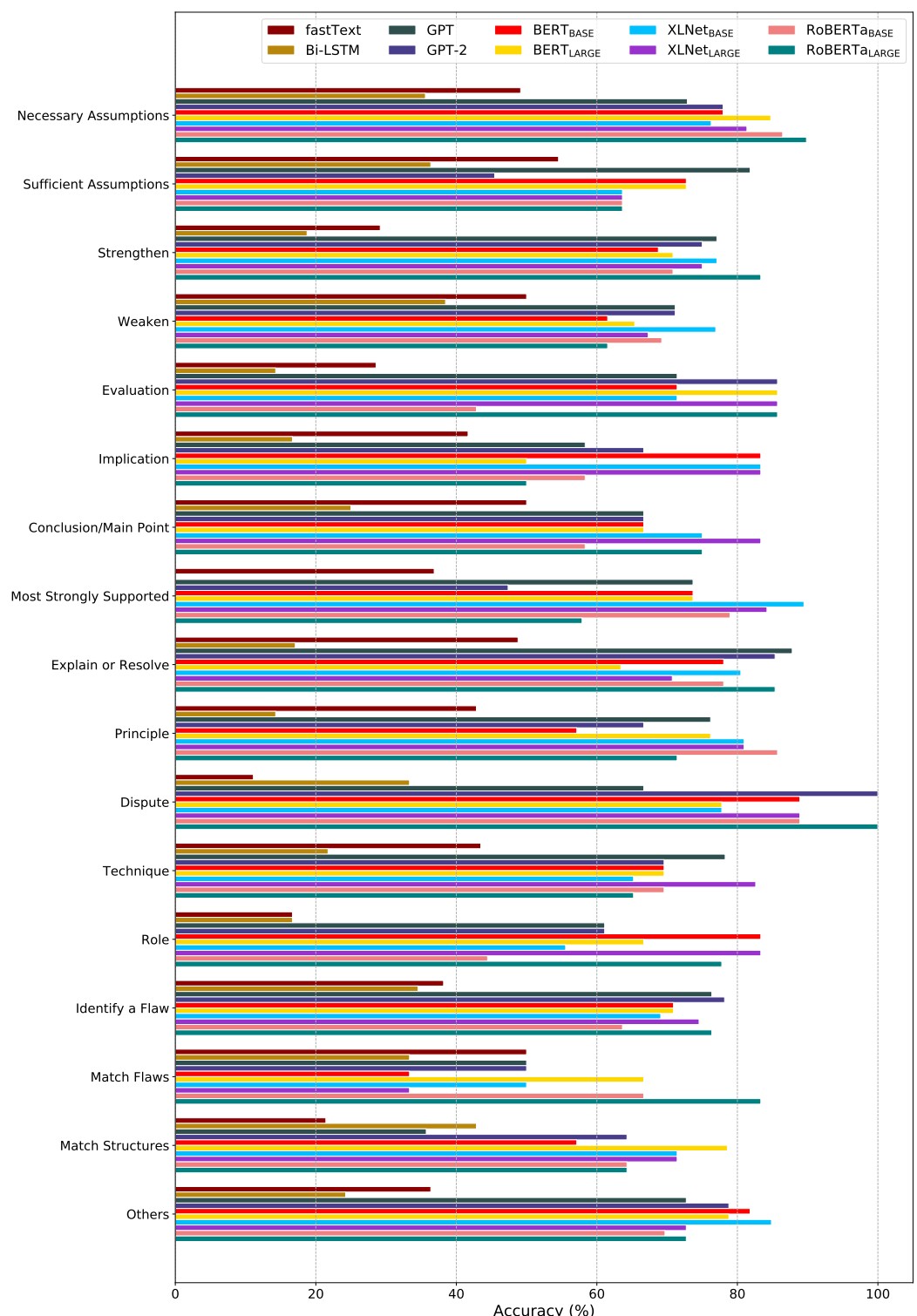

Figure 6: Accuracy of all baseline models on EASY set of testing set

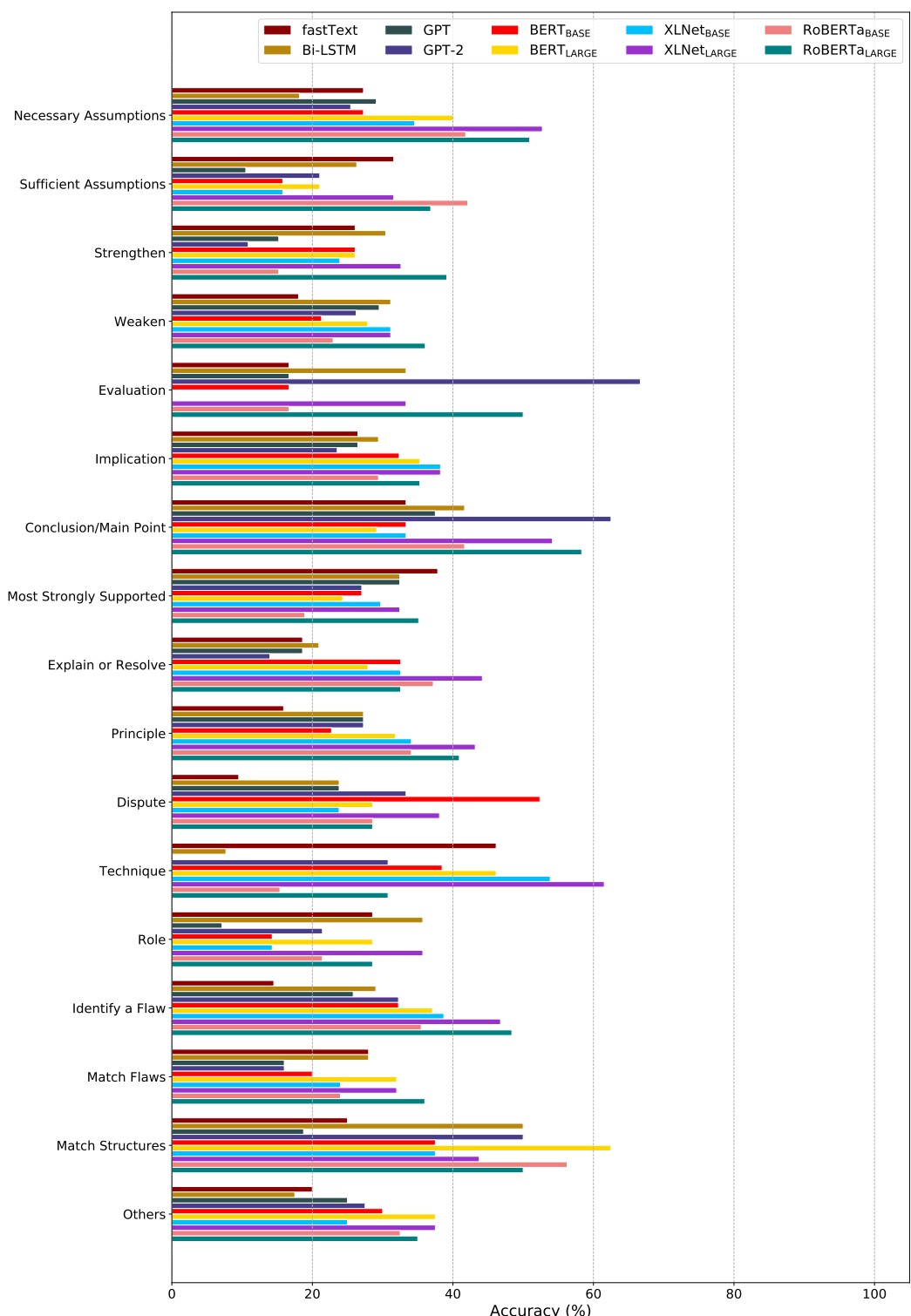

Figure 7: Accuracy of all baseline models on HARD set of testing set

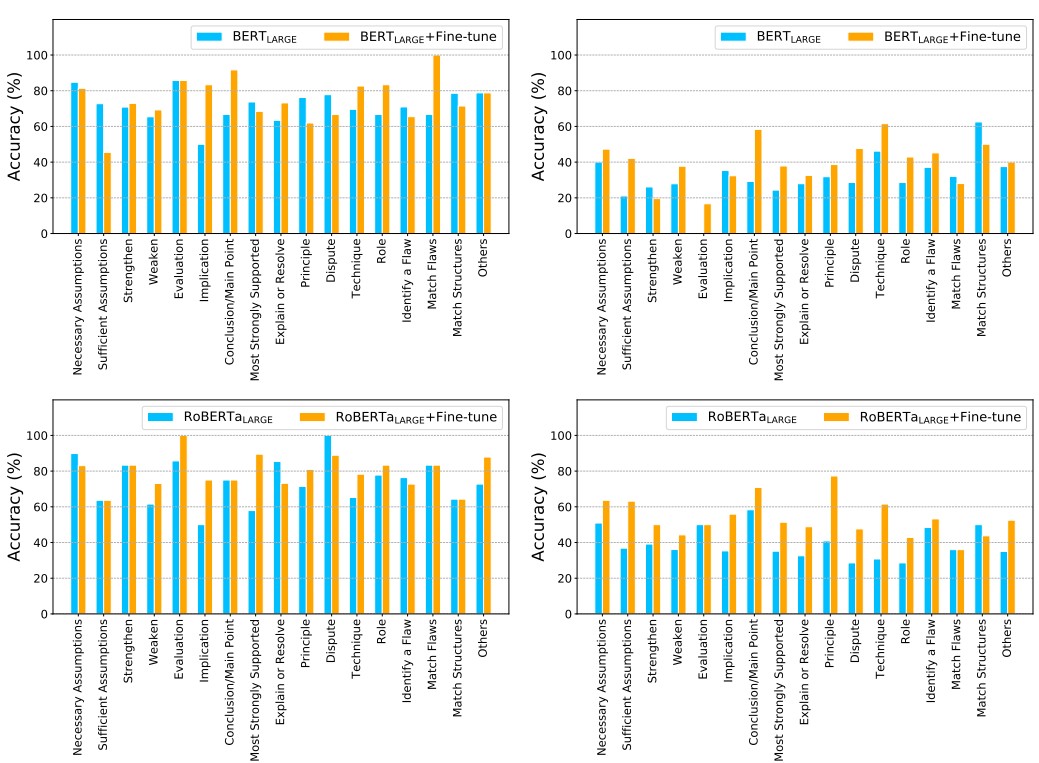

Figure 8: Performance of BERT$_{\text{LARGE}}$ (top) and RoBERTa$_{\text{LARGE}}$ (bottom) on EASY (left) and HARD (right) testing sets.

