# OpenReview forum: "ReClor: A Reading Comprehension Dataset Requiring Logical Reasoning"
_ICLR.cc/2020/Conference — Accept (Poster)_

### Official Review · AnonReviewer1 · 2019-10-10
**Official Blind Review #1**

**Rating:** 8

**Review:**

This paper presents a small multiple choice reading comprehension dataset drawn from LSAT and GMAT exams.  I like the idea of using these standardized tests as benchmarks for machine reading, and I think this will be a valuable resource for the community.

The two major concerns I have with this paper are with its presentation and with the quality of the baselines.  These concerns leave me at a weak accept, instead of a strong accept, which I would be if these issues were fixed.

Presentation:

The main contribution here is in collecting a new dataset drawn from the LSAT and GMAT.  (I assume that is all that is used, though the text actually says "such as".  Or is this from practice exams and not the actual exams?)  The job of this paper is to convince me, a researcher focusing on reading comprehension, that I should use this dataset as a target of my research.  Out of 8 pages, however, at most 2 are devoted to actually describing this contribution and why it's a good target for reading comprehension.  Much of the interesting description of the phenomena in the dataset is relegated to the appendix, which I did not really look at because it far exceeds the page limit of an ICLR submission.  I have a lot of questions about this dataset that could have been answered in the main text had more of the paper been given to actually describing the data.  Such as:

- What are some actual examples of the different kinds of reasoning in table 2?
- How did you decide the proportions of questions listed in table 2?  Was that manual?
- Where did the questions actually come from?  Real exams?  Practice exams?  Which ones?
- What is the reasoning process that a person would actually use to answer some of these questions?  Preferably answered for several of the question types that you listed.  You have 8 pages; there's a lot of space that could be given to this.
- What kinds of things do you see in the distractors that make this hard?

I'm recommending a weak accept for this paper, but only because of my background knowledge about the LSAT and GMAT, and my prior understanding of reading comprehension and why this would be a really interesting dataset.  I think the paper itself could do a *much* better job convincing a reader about the dataset than what is done here.

What should be cut to make room for this?

I think far too much space is given to the discussion of easy vs. hard (~2.5 pages).  Given that the best performance is at ~54% on the full test set, there isn't a lot of need for this.  The argument about four flips of a 25% coin giving a 0.39% chance of guessing right every time is only true if the coin flips are independently drawn; training the same pretrained model on the same data with a different random seed is clearly not independent, so this argument rings hollow.  It's not really clear what to conclude from the easy vs. hard split, other than the models that you used to create it expectedly do well on the easy split and hard on the test split.  You'd want to have separate models that create the split than what you're evaluating them with, but even then, you're training them on the same data, so it's still not really clear what to conclude.  It's sufficient in a case like this to just evaluate as a baseline a model that shouldn't have access to enough information to solve the task, and measure its performance on the test set.  That would have taken a few sentences to describe, instead of 2.5 pages.

You could also recover a lot of space by evaluating fewer baselines.  Describing six baseline models and including all of their performance numbers takes up a lot of space, and we really don't need to know what all of those numbers are.  One or two, along with question-only and option-only baselines, would be plenty.  If you really want to include all of them, that's information that should go into the appendix, instead of something that's core to your paper's contribution, like describing the dataset that you're releasing.

Quality of baselines:

With a dataset this small, you definitely want to pretrain BERT / RoBERTa on a larger QA dataset before training on ReClor.  I would guess that performance would be significantly higher if you used a RACE-trained model as a starting point, instead of raw BERT / RoBERTa.  And if it doesn't end up helping, then you've made a stronger argument about how your dataset is different from what has come before, and a good target for future research.  What I would recommend for table 6:

- chance
- option-only
- question-only
- raw XLNet or RoBERTa
- XLNet or RoBERTa pretrained on RACE, or SQuAD, or similar

And I would pick either XLNet or RoBERTa based on which one did the best after pre-training.  That way, related to my first point, you can remove the description of all other models, and table 4, and free up a bunch of space for more interesting things.

As an aside, I think it's good that the dataset is relatively small, as it gives these over-parameterized models less opportunity to overfit, and it makes us come up with other ways of obtaining supervision.  But you've got to be sure to use the best baselines available, and with small datasets, that means more pretraining on other things.

EDIT Nov. 19, 2019: The authors' revision has satisfied my concerns.  I think that this is a good paper and it should be accepted.

**Experience Assessment:**

I have published in this field for several years.

**Review Assessment: Checking Correctness Of Derivations And Theory:**

N/A

**Review Assessment: Checking Correctness Of Experiments:**

I carefully checked the experiments.

**Review Assessment: Thoroughness In Paper Reading:**

I read the paper thoroughly.

---

> ### Author Response · Authors · 2019-11-14
> **Response (Part 1/2)**
>
> Thank you very much for your valuable and detailed suggestions about the presentation and quality of baselines! We have revised our paper according to your suggestions
>
>
> #Q1: What are some actual examples of the different kinds of reasoning in table 2?
> #A1: We have added some examples in Figure 2 in the revised paper.
>
>
> #Q2: How did you decide the proportions of questions listed in table 2? Was that manual?
> #A2: We collect questions from the logical reasoning module of real exams (91.22%) of LSAT, GMAT and practice exams (8.78%). The proportions reflect those of logical reasoning question types in LSAT and GMAT.
>
>
> #Q3: Where did the questions actually come from? Real exams? Practice exams? Which ones?
> #A3: Most of the questions (91.22%) of this dataset are from actual LSAT and GMAT exams, and the other (8.78%) are from high-quality practice exams.
>
>
> #Q4: What is the reasoning process that a person would actually use to answer some of these Questions?
> #A4:  We have added examples about how humans would solve the questions in Table 3.
>
>
> #Q5: The argument about four flips of a 25% coin giving a 0.39% chance of guessing right every time is only true if the coin flips are independently drawn; training the same pretrained model on the same data with a different random seed is clearly not independent, so this argument rings hollow.  It's not really clear what to conclude from the easy vs. hard split, other than the models that you used to create it expectedly do well on the easy split and hard on the test split.  You'd want to have separate models that create the split than what you're evaluating them with, but even then, you're training them on the same data, so it's still not really clear what to conclude.  It's sufficient in a case like this to just evaluate as a baseline a model that shouldn't have access to enough information to solve the task, and measure its performance on the test set.
> #A5: We admit this method can only roughly split the testing set, but it can still provide us with some insights.
> 1. As shown in Table 8, simple model fastText can achieve accuracy as high as 40.2% in the easy testing set, indicating that some of the problems could be easily solved by calculating the statistical information at word-level/n-gram level in context, question and answers without truly understanding the logical relationship. One the other hand, the performance of fastText on the hard set is exactly around 25% like random guess, showing that these questions are relatively harder and need further logical reasoning and understanding ability to solve.
> 2. We also find that if we give strong assumption that BERT_base can steadily predict x data points when only answer options is input,  and the other data point (1000-x) could only be random guessed (25%), we can obtain an equation x+(1000-x)*0.25=432 ( performance of BERT_base with only options input on ReClor testing set is 43.2%). Solve this equation, x=~243, near the number of filtered data, ie. 234.  This calculation indicates the split is reasonable to some extent.
> 3. As suggested, we exclude XLNet and use only 4 models (i.e. GPT, GPT2, BERT, RoBERTa) to form the EASY set. The size of the new EASY set is 410, which is comparable to the original size (440). We further evaluate XLNet_Large on the new split, and get the accuracy of 72.9/40.0 on EASY/HARD set, which is also similar to the original accuracy 72.5/38.6.
> 4. Splitting out biased set and non-biased set can be more straightforward and intuitively reflect different models’ abilities in capturing bias and performing true logical reasoning (Sugawara et al., 2018). Although this split method is heuristic, we hope it still remains in the paper and inspires more refined methods to be proposed.
> Reference:
> Saku Sugawara, Kentaro Inui, Satoshi Sekine, and Akiko Aizawa. What makes reading comprehension questions easier? In Proceedings of the 2018 Conference on Empirical Methods in Natural Language Processing, pp. 4208–4219, 2018.
>
> Continued in "Response (Part 2/2)"

---

> > ### Author Response · Authors · 2019-11-14
> > **Response (Part 2/2)**
> >
> > #Q6: What is the reasoning process that a person would actually use to answer some of these questions?  Preferably answered for several of the question types that you listed.  You have 8 pages; there's a lot of space that could be given to this. What kinds of things do you see in the distractors that make this hard?
> > #A6: We provide this content in our revised Table 4.
> >
> >
> > #Q7: With a dataset this small, you definitely want to pretrain BERT / RoBERTa on a larger QA dataset before training on ReClor.  I would guess that performance would be significantly higher if you used a RACE-trained model as a starting point, instead of raw BERT / RoBERTa.  And if it doesn't end up helping, then you've made a stronger argument about how your dataset is different from what has come before, and a good target for future research.  What I would recommend for table 6: - chance - option-only - question-only - raw XLNet or RoBERTa - XLNet or RoBERTa pretrained on RACE, or SQuAD, or similar
> > #A7: Thanks for your helpful and effective suggestion. This method indeed helps. We have added more experimental results in Table 7. According to the results, fine-tuning on RACE first can bring significant improvement. For example, XLNet_large improves overall accuracy from 53.5% to 62.7% (but it is still far away from ceiling performance). This observation is consistent with what has been reported in recent literature.  Jin et al. (2019) find that by first fine-tuning on RACE and then further fine-tuning on the target dataset, the performances of BERT_base on multiple-choice dataset MC500 (Richardson et al., 2013) and DREAM (Sun et al., 2019) can significantly boost from 69.5% to 81.2%, and from 63.2% to 70.2%, respectively.
> > The histograms in the bottom of Figure 4 show that on EASY set, accuracy of XLNet_large improves a similar amount among different question type, while on HARD set, we can see a significant improvement on some question types such as Evaluation, Conclusion and Summary/Main Point. This may be because these types require less logical reasoning to some extent compared with other types, and similar question types may also be found in RACE dataset. Thus, the pre-training on RACE helps enhance the ability of logical reasoning especially of relatively simple reasoning types, but more methods are still needed to further enhance the ability especially that of relative complex reasoning types.
> > References:
> > Di Jin, Shuyang Gao, Jiun-Yu Kao, Tagyoung Chung, and Dilek Hakkani-tur.  Mmm:  Multi-stage multi-task learning for multi-choice reading comprehension.arXiv preprint arXiv:1910.00458,2019.
> > Matthew Richardson, Christopher JC Burges, and Erin Renshaw.  Mctest:  A challenge dataset for the open-domain machine comprehension of text.   InProceedings of the 2013 Conference onEmpirical Methods in Natural Language Processing, pp. 193–203, 2013.
> > Kai Sun, Dian Yu, Jianshu Chen, Dong Yu, Yejin Choi, and Claire Cardie. Dream: A challenge dataset and models for dialogue-based reading comprehension.Transactions of the Association forComputational Linguistics, 7:217–231, 2019.

---

### Official Review · AnonReviewer3 · 2019-10-21
**Official Blind Review #3**

**Rating:** 6

**Review:**


Paper Summary:

This paper presents a machine reading comprehension dataset called ReClor. It is different from existing datasets in that ReClor targets logical reasoning. The authors identified biased data points and separated the testing dataset into biased and non-biased sets. Experimental results show that state-of-the-art models such as XLNet and RoBERTa struggle on the non-biased HARD set with poor performance near that of random guess.

Strengths:

—The dataset, which is extracted from standardized tests such as GMAT and LSAT, requires the ability to perform complex logical reasoning. It is difficult for crowdsourcing workers to generate such logical questions.

—The authors carefully analyzed the biases, which are often exploited by models to achieve high accuracy without truly understanding the text.

—The authors thoroughly investigated how well strong models such as RoBERTa can perform.

—The paper is well organized and well written.

Weaknesses:

—The dataset seems small to acquire the ability to perform complex logical reasoning. The training, validation, and testing datasets consist of 4,651, 500, and 1,000 questions, respectively.

—The paper does not show statistics of the dataset such as question/passage length and question vocabulary size.

—The paper does not show results of models trained with other reading comprehension datasets such as RACE and fine-tuned with ReClor.

—Unlike other datasets, the questions themselves show their required logical reasoning types in ReClor. For example, the question “Which one of the following is an assumption required by the argument?” shows the “Necessary Assumptions” type and has no information with respect to passages. This characteristic makes it difficult to use the ReClor dataset as an evaluation benchmark for models trained with large-scaled reading comprehension datasets such as RACE.

—The human performance with respect to different question types of logical reasoning is not analyzed in Figure 3.

Review Summary:

The paper is well motivated. ReClor can be a useful dataset to evaluate the ability of logical reasoning, while the dataset seems small to acquire the ability to perform complex logical reasoning.   I think it can benefit a lot with a more comprehensive analysis of transfer learning with other reading comprehension datasets.

EDIT Nov. 20, 2019:
I appreciate the authors' revision.
Although my concern about the size of the dataset is not satisfied, I decided to increase the score of the paper (weak reject -> weak accept) upon looking at the author response about transfer learning.

There is a typographical error, in Table 2.  ACR -> ARC.

**Experience Assessment:**

I have published one or two papers in this area.

**Review Assessment: Checking Correctness Of Derivations And Theory:**

N/A

**Review Assessment: Checking Correctness Of Experiments:**

I assessed the sensibility of the experiments.

**Review Assessment: Thoroughness In Paper Reading:**

I read the paper at least twice and used my best judgement in assessing the paper.

---

> ### Author Response · Authors · 2019-11-14
> **Response**
>
> Thank you very much for your valuable and detailed comments!
>
>
> #Q1: The dataset seems small to acquire the ability to perform complex logical reasoning. The training, validation and testing datasets consist of 4,651, 500, and 1,000 questions, respectively.
> #A1: High-quality logical reasoning questions from exams are very hard to collect. We try our best to collect 6K+ questions, and this size is comparable to the recent two exam datasets ARC and DREAM.
> Reference:
> [ARC] Peter Clark, Isaac Cowhey, Oren Etzioni, Tushar Khot, Ashish Sabharwal, Carissa
> Schoenick, and Oyvind Tafjord. Think you have solved question answering? try arc, the ai2
> reasoning challenge.arXiv preprint arXiv:1803.05457, 2018.
> [DREAM] Kai Sun, Dian Yu, Jianshu Chen, Dong Yu, Yejin Choi, and Claire Cardie. Dream: A challenge dataset and models for dialogue-based reading comprehension.Transactions of the Association for Computational Linguistics, 7:217–231, 2019.
>
>
> #Q2: The paper does not show statistics of the dataset such as question/passage length and question vocabulary size.
> #A2: Thanks for your input. We show more statistical information about ReClor in Table 2 in the revised version.
>
>
> #Q3: The paper does not show results of models trained with other reading comprehension datasets such as RACE and fine-tuned with ReClor.
> #A3: Thanks for your helpful and effective suggestion. This method does work. We have added more experimental results in Table 7. According to the results, fine-tuning on RACE first can indeed bring significant improvement. For example, XLNet_large improves overall accuracy from 53.5% to 62.7% (but it is still far away from ceiling performance). This observation is consistent with what has been reported in recent literature.  Jin et al. (2019) find that by first fine-tuning on RACE and then further fine-tuning on the target dataset, the performances of BERT_base on multiple-choice dataset MC500 (Richardson et al., 2013) and DREAM (Sun et al., 2019) can significantly boost from 69.5% to 81.2%, and from 63.2% to 70.2%, respectively.
> The histograms at the bottom of Figure 4 show that on EASY set, accuracy of XLNet_large improves a similar amount among different question types, while on HARD set, we can see a significant improvement on some question types such as Evaluation, Conclusion and Summary/Main Point. This may be because these types require less logical reasoning to some extent compared with other types, and similar question types may also be found in RACE dataset. Thus, the pre-training on RACE helps enhance the ability of logical reasoning especially of relatively simple reasoning types, but more methods are still needed to further enhance the ability especially that of relative complex reasoning types.
>
> References:
> Di Jin, Shuyang Gao, Jiun-Yu Kao, Tagyoung Chung, and Dilek Hakkani-tur.  Mmm:  Multi-stage multi-task learning for multi-choice reading comprehension.arXiv preprint arXiv:1910.00458,2019.
> Matthew Richardson, Christopher JC Burges, and Erin Renshaw.  Mctest:  A challenge dataset for the open-domain machine comprehension of text.   InProceedings of the 2013 Conference on Empirical Methods in Natural Language Processing, pp. 193–203, 2013.
> Kai Sun, Dian Yu, Jianshu Chen, Dong Yu, Yejin Choi, and Claire Cardie. Dream: A challenge dataset and models for dialogue-based reading comprehension.Transactions of the Association for Computational Linguistics, 7:217–231, 2019.
>
>
> #Q4: Unlike other datasets, the questions themselves show their required logical reasoning types in ReClor. For example, the question “Which one of the following is an assumption required by the argument?” shows the “Necessary Assumptions” type and has no information with respect to passages. This characteristic makes it difficult to use the ReClor dataset as an evaluation benchmark for models trained with large-scaled reading comprehension datasets such as RACE.
> #A4: As you suggest, we added section 4.3 Transfer Learning Through Fine-tuning. The models are first trained on RACE before further fine-tuned on ReClor. The results are reported in Table 7. In this setting, different models have different performances on ReClor. Thus, ReClor can serve as an evaluation benchmark for models trained with RACE in this sequential fine-tuning setting.
>
>
> #Q5: The human performance with respect to different question types of logical reasoning is not analyzed in Figure 3.
> #A5: Since we only randomly choose 100 samples from testing set to test human performance, some question types are not included in the testing set, therefore we choose to only provide the overall accuracy.

---

### Official Review · AnonReviewer2 · 2019-10-21
**Official Blind Review #2**

**Rating:** 6

**Review:**

This paper presents a new reading comprehension dataset for logical reasoning. It is a multi-choice problem where questions are mainly from GMAT and LSAT, containing 4139 data points. The analyses of the data demonstrate that questions require diverse types of reasoning such as finding necessary/sufficient assumptions, whether statements strengthen/weaken the argument or explain/resolve the situation. The paper includes comprehensive experiments with baselines to identify bias in the dataset, where the answer-options-only model achieves near half (random is 25%). Based on this result, the test set is split into the easy and hard set, which will help better evaluation of the future models. The paper also reports the numbers on the split data using competitive baselines where the models achieve low performance on the hard set.

Strengths
1) The introduced dataset is timely and is more challenging than existing reading comprehension datasets, based on the examples presented in the paper.
2) Analyses of reasoning types (Table 2) are done in a very comprehensive way, especially because they are comparable with descriptions from previous literature. This analysis clearly demonstrates that this data requires diverse types of challenging reasoning.
3) The experiments are comprehensive with many competitive baseline models, and their efforts to identify bias and use the result to split the test set are impressive.

Weaknesses
1) Although the main claim of this paper is about logical reasoning, I believe the term “logical reasoning” is somewhat subjective. What exactly is the definition of logical reasoning? The paper never defines it in any way; it only shows Table 1 as an example.
2) (Continuing the point above) The paper mentions other datasets’ statistics on logical reasoning (in Section 1, they mention “0% in MCTest dataset and 1.2% SQuAD” requires logical reasoning). How was this analysis done?
3) (Continuing the point above) Are all types of reasoning in Table 2 logical reasoning? I agree that those reasoning are very challenging, but I am not convinced why some of them belong to logical reasoning (E.g. summary). I expect this problem can be resolved if authors bring a clearer definition of logical reasoning.
4) Although I appreciate the experiments to identify bias, I believe that the bias seems to be somewhat significant if 440 out of 1000 examples are identified as biased examples. I believe that the authors should do more comprehensive analysis on biased examples and explain what are possible biases here, in order to justify the significant bias.

Marginal comments
1) The authors claim they collected the questions from open websites and books, but they didn’t provide the URLs or names. More information about which source was used, how data points were filtered and so on would be necessary to assess the quality of the dataset. Specifically, is there any particular reason for removing one of wrong options to make it have four choices, instead of five choices?
2) Why is the number of Chance model 0.39? Shouldn’t it be 1000 * 0.39% = 3.9?
3) In Table 5, it’s interesting to see that different models identify different numbers of biased examples. Have you looked at how much do they overlap each other if you take a pair of models? (A number of the union kinda shows it, but I wonder if there is a more intuitive way to see how consistent the models’ decisions are.)
4) It would be also helpful to see the baseline results of “question & answer options only model” and “context & answer options only model”.

**Experience Assessment:**

I have published one or two papers in this area.

**Review Assessment: Checking Correctness Of Derivations And Theory:**

I assessed the sensibility of the derivations and theory.

**Review Assessment: Checking Correctness Of Experiments:**

I assessed the sensibility of the experiments.

**Review Assessment: Thoroughness In Paper Reading:**

I read the paper at least twice and used my best judgement in assessing the paper.

---

> ### Author Response · Authors · 2019-11-14
> **Response**
>
> Thank you very much for your valuable and detailed comments!
>
>
> #Q1: What exactly is the definition of logical reasoning?
> #A1: In the second paragraph of the Introduction of the original version, we cited the
> definition from Law School Admission Council (LSAC) — examine, analyze and critically
> evaluate arguments as they occur in ordinary language [Link1]. We highlight this definition in the new version of the paper.
> Reference:
> [Link1] https://www.lsac.org/lsat/taking-lsat/test-format/logical-reasoning
>
>
> #Q2: The paper mentions other datasets’ statistics on logical reasoning (in Section 1, they mention “0% in MCTest dataset and 1.2% SQuAD” requires logical reasoning). How was this analysis done?
> #A2: This analysis was done by Sugawara & Aizawa (2016) (from their Table 2).
> Reference:
> Saku Sugawara and Akiko Aizawa. An analysis of prerequisite skills for reading
> comprehension. In Proceedings of the Workshop on Uphill Battles in Language Processing:
> Scaling Early Achievements to Robust Methods, pp. 1–5, 2016.
>
>
> #Q3: Are all types of reasoning in Table 2 logical reasoning? I agree that those reasoning are very challenging, but I am not convinced why some of them belong to logical reasoning (E.g. summary).
> #A3: The definition of reasoning type of Summary/Main Point in our paper is to identify the **main point** of **an argument**, which requires to disentangle different parts of an argument and recognize their main part/point according to their logical relationships. Thus, this type of reasoning meets the definition of logical reasoning from Law School Admission Council (LSAC). LSAC also includes this type of questions in the logical reasoning module of LSAT.
>
>
> #Q4: Although I appreciate the experiments to identify bias, I believe that the bias seems to be somewhat significant if 440 out of 1000 examples are identified as biased examples. I believe that the authors should do more comprehensive analysis on biased examples and explain what are possible biases here, in order to justify the significant bias.
> #A4: We admit that the split method proposed in the paper can only roughly split the
> testing set. Below are some possible justifications of the biases: option length and word-level cues.
> 1. The average length of right/wrong options in EASY set are 23.5/21.6, while those in HARD set are 23.0/22.4. This indicates that option sentence length may be one of the accountable biases.
> 2. FastText is a word-embedding based model. As shown in Table 7, there is a significant performance gap between EASY (biased) set and HARD (non-biased) set, i.e., 40.2% and 23.4% respectively. The relatively high performance on EASY set indicates that some of the problems could be easily solved by calculating the statistical information in word-level/n-gram level about context, question and answers without truly understanding the logical relationship. However, the performance of fastText on the HARD set is exactly around 25% like random guess, showing that these questions are relatively harder and need further logical reasoning and understanding ability to solve. Therefore, word-level cues may be another bias.
>
>
> #Q5: The authors claim they collected the questions from open websites and books, but they didn’t provide the URLs or names. More information about which source was used, how data points were filtered and so on would be necessary to assess the quality of the dataset. Specifically, is there any particular reason for removing one of wrong options to make it have four choices, instead of five choices?
> #A5: One of the sources is [link2]. We will list down all the sources used for data collection in the dataset.
> We manually filtered the data points by ensuring: 1) they are from real exams of GMAT and LSAT (91.22%), or high-quality practice exam for the GMAT and LSAT (8.78%); 2) they belong to logical reasoning module of these exams; 3) they are unambiguous.
> As explained in section 3.1, we remove one of the choices to comply with fair use of law [link3].
> References:
> [link2] https://www.lsac.org/lsat/taking-lsat/test-format/logical-reasoning/logical-reasoning-sample-questions
> [link3] https://www.copyright.gov/fair-use/more-info.html
>
>
> #Q6: Why is the number of Chance model 0.39? Shouldn’t it be 1000 * 0.39% = 3.9?
> #A6: Thank you again for pointing it out. We've modified it in the revised version.
>
>
> #Q7: In Table 5, it’s interesting to see that different models identify different numbers of biased examples. Have you looked at how much do they overlap each other if you take a pair of models?
> #A7: The average of a single model's decision (among the intersection of 4 random seed) is 228, while the average of the overlap of two models is 138. We include the details in Table 28 in the Appendix.
>
>
> #Q8: It would be also helpful to see the baseline results of “question & answer options only model” and “context & answer options only model”.
> #A8: The supplementary results are listed in Table 7 in the revised version.

---

### Author Response · Authors · 2019-11-14
**General ﻿Response**

Dear Reviewers,

Thanks a lot for reviewing our paper and for your valuable comments.
To incorporate your feedback, we have uploaded a revision of our paper with the following changes:

1. Add Table 2 “Statistics of several multiple-choice MRC datasets” including ReClor.
2. Add Table 4 “Two examples to show how humans to solve the questions”.
3. Add Figure 2 “Examples of some question types”.
4. Add Section 4.3 to describe the fine-tuning setting for transfer learning.
5. Extend Table 7 to include experiment results of fine-tuning and different input settings.
6. Extend Figure 4 to include results of fine-tuning with respect to different question types.
7. Description of baseline models and implementation are moved to the Appendix A and B due to the limit of the room.

---

### Decision · Program_Chairs · 2019-12-19

**Decision:**

Accept (Poster)

**Comment:**

Main content:

Blind review #1 summarizes it well:

This paper presents a new reading comprehension dataset for logical reasoning. It is a multi-choice problem where questions are mainly from GMAT and LSAT, containing 4139 data points. The analyses of the data demonstrate that questions require diverse types of reasoning such as finding necessary/sufficient assumptions, whether statements strengthen/weaken the argument or explain/resolve the situation. The paper includes comprehensive experiments with baselines to identify bias in the dataset, where the answer-options-only model achieves near half (random is 25%). Based on this result, the test set is split into the easy and hard set, which will help better evaluation of the future models. The paper also reports the numbers on the split data using competitive baselines where the models achieve low performance on the hard set.

--

Discussion:

While the authors agree this is an important direction, there are reservations concerning the small size of the dataset, that have not been fully addressed.

--

Recommendation and justifcation:

I still believe this paper should be accepted as the existing datasets for reading comprehension are inadequate and it is important for the field not to be climbing the wrong hill.